# Adipokines and Endotoxemia Correlate with Hepatic Steatosis in Non-Alcoholic Fatty Liver Disease (NAFLD)

**DOI:** 10.3390/nu12030699

**Published:** 2020-03-05

**Authors:** Anika Nier, Yvonne Huber, Christian Labenz, Maurice Michel, Ina Bergheim, Jörn M. Schattenberg

**Affiliations:** 1Department of Nutritional Sciences, Molecular Nutritional Science, University of Vienna, 1090 Vienna, Austriaina.bergheim@univie.ac.at (I.B.); 2Metabolic Liver Research Program, I. Department of Medicine, University Medical Centre of the Johannes Gutenberg-University, Langenbeckstrasse 1, 55131 Mainz, Germany

**Keywords:** non-alcoholic fatty liver disease, dietary fiber consumption, bacterial endotoxin, hepatic fibrosis

## Abstract

(1) Background: The etiology of non-alcoholic fatty liver disease (NAFLD) is multifactorial. Dietary composition has been implicated as a factor modulating intestinal barrier and could affect disease severity. The aim of this study was to evaluate dietary intake and markers of intestinal permeability in patients with NAFLD. (2) Methods: We enrolled 63 patients with NAFLD and compared them to age-matched controls. (3) Results: body mass index (BMI) and leptin to adiponectin ratio—the latter being an indicator of abdominal fat accumulation—correlated with the degree of hepatic steatosis being accompanied with rising levels of fasting insulin. Furthermore, endotoxin plasma levels and markers of inflammation were significantly higher in NAFLD compared to controls and increased with the severity of hepatic steatosis. Despite comparable intake of total energy and macronutrients, intake of fiber was lower in all patients with NAFLD compared to controls and were negatively related to disease severity. (4) Conclusions: Taken together, results of the present study suggest that fiber intake in patients is negatively related to steatosis degree and bacterial endotoxin levels, further suggesting that dietary fiber intake may be a target in NAFLD treatment (NCT: 02366052 and 03482284).

## 1. Introduction

Non-alcoholic fatty liver disease (NAFLD) has become the most frequent liver disease in Europe with an estimated prevalence of 24% [1,2]. In the U.S. it is the leading cause for liver transplantation in women [3]. The rise in prevalence has translated into a growing economic and social burden and further increases in health care expenditures are expected [1]. NAFLD comprises a wide spectrum of increasing severity ranging from simple steatosis to steatohepatitis (non-alcoholic steatohepatitis; NASH), fibrosis, cirrhosis or even hepatocellular carcinoma [4]. Importantly, NAFLD is an independent contributor to overall and liver-specific mortality [5].

The pathophysiology of NAFLD involves hepatic necroinflammation that develops in the context of hepatic steatosis, lipotoxicity and insulin resistance [6]. Besides the genetic background, external factors including lifestyle and nutrition are major determinants of the development of NAFLD and may explain regional differences in disease prevalence [7]. Currently, no approved pharmacological treatment is available and the first line recommendations are changes in lifestyle by increasing energy expenditure and decreasing energy uptake [6,8]. These recommendations are difficult to implement for patients [9], and in particular complex lifestyle interventions are afflicted by low compliance rates [10].

The pathophysiology of NAFLD involves multiple, partly redundant mechanisms that lead to the initiation of lobular inflammation fostering disease progression in a subset of patients. Among the factors that have been identified in animal models is the transition of endotoxin likely resulting from impaired intestinal permeability and influenced by metagenomic richness [11]. This is critically influenced by the dietary pattern and in particular the dietary fiber content [12]. Nutritional fiber can affect the rate of nutrient resorption, gut mortality, and thus affect the intestinal barrier in general. Epidemiological data from a multi-ethnic cohort [13] and findings in animal models [14] suggest that a low intake of fiber contributes to the development of NAFLD. Indeed, a landmark study further highlighted the fact that dietary intake of fiber is markedly lower in overweight patients with NAFLD when compared to isocaloric nourished weight matched individuals without NAFLD [15]. Moreover, we have previously shown that a diet low in fiber is associated with higher endotoxin levels and higher alanine aminotransferase (ALT) levels in normal weight healthy subjects [16]. However, studies addressing endotoxin plasma levels in relation to dietary fiber intake and markers for visceral fat accumulation in patients with NAFLD are still lacking. Further limitations to these findings arise from the lack of well-controlled, prospective trials performed by hepatologists and nutritional scientists to stage patients with NAFLD. Indeed, to our knowledge, previous studies assessing intestinal permeability, abdominal fat accumulation and nutritional intake in patients with NAFLD did not stratify patients for degree of steatosis. Therefore, the aim to of this study was to assess differences in disease severity in patients with NAFLD within a prospective study protocol and compare those of healthy controls with dietary-particular focus on macronutrient and fiber intake as well as markers for visceral fat accumulation and intestinal permeability. Additionally, exploration of experimental markers of disease progression and inflammation were assessed.

## 2. Materials and Methods

### 2.1. Study Participants and Study Design

Patients and healthy volunteers were recruited at the outpatient hepatology clinic of the University Medical Center of the Johannes Gutenberg-University Mainz, Germany and at the Department of Nutritional Sciences, Vienna, Austria. In total, 63 patients with NAFLD and 14 healthy normal weight controls were enrolled in the present study. Patients with NAFLD were enrolled within the prospective Nutritional Counselling vs. Nutritional Supplements for NASH—a Randomized Prospective, Open Label Pilot Study (NUCES NASH study; clinical trials.gov NCT02366052). Healthy controls were enrolled within in the so-called ENDO-META study (Effect of Monosaccharides on Intestinal Barrier Function, clinical trials.gov NCT 03482284). As liver histology was not available in all patients, we use the term NAFLD to describe the study population throughout this manuscript. However, NASH was defined by liver histology in a subset of patients. In patients without liver histology, we required the presence of hepatic steatosis and elevated cytokeratin 18 fragments (M30 > 200 µg/mL) as an indicator of disease activity [17] and as such the study population is enriched with patients suffering from NASH. The degree of hepatic steatosis was stratified into mild (grade 1; *n* = 20), moderate (grade 2; *n* = 31) or severe (grade 3, *n* = 12) steatosis according to published semiquantitative scoring systems in a standardized manner by a trained study physician [18]. Healthy volunteers underwent standard medical assessment including abdominal ultrasound and laboratory assessment. In this group liver disease, arterial hypertension and diabetes mellitus were ruled-out and they were subsequently considered as a control group (*n* = 14). Alcohol consumption was assessed by the study physician and nutritionists and participants exceeding 20 g/day for female and 30 g/day for male were not considered NAFLD according to current guidelines [19]. A complete medical history, anthropometric data, and nutritional intake through assessment of a trained nutritionist were obtained. Laboratory assessments consisted of complete blood count, parameters of liver function, creatinine and experimental labs including cytokines and experimental markers of intestinal permeability, as well as metabolic and inflammatory markers in participants.

### 2.2. Dietary Assessment, Anthropometric Data and Blood Pressure

Nutritional intake was assessed by two independently performed 24 h recalls conducted by a trained nutritionist. The software EBIS-Pro^©^, Version 2011 (Dr. Jürgen Erhardt, Willstätt-Legelshurst, Germany) was used to analyze nutritional intake data. Two participants in the control group and one participant in the steatosis grade 1 group were excluded from the nutritional analysis due to incompliance in dietary assessment. Weight, height and blood pressure was assessed in all participants at the study centers.

### 2.3. Blood Sampling and Laboratory Measurements

Fasting blood (8 h) samples were collected. Standard labs were analyzed in the clinical routine laboratory. Additionally, leptin (Hölzel GmbH, Wildberg, Germany), plasminogen activator inhibitor (PAI)-1 (LOXO GmbH, Germany), lipopolysaccharide-binding protein (LBP; Abnova, Taipei City, Taiwan), c-reactive protein (CRP; DRG Instruments GmbH, Marburg, Germany) and adiponectin (TECOmedical AG, Sissach, Switzerland) were determined by commercially available ELISA kits according to the manufacturer’s instructions.

### 2.4. Endotoxin Measurement

Endotoxin measurements have been standardized [20] and were determined in plasma samples after heating for 20 min at 70 °C. Tween^®^ 80 (Carl Roth GmbH + Co. KG, Karlsruhe, Germany) was added (20%) and samples were treated with ultrasound for 5 min. Endotoxin levels were measured using a commercially available endpoint enzymatic assay kit (Charles River, Germany) based on *limulus amebocyte lysate*. Recovery rates were 104% on average.

### 2.5. Statistical Analysis

Results are presented as absolute numbers or as mean ± standard error of the mean (SEM). Log-transformation of data was performed to approach normal distribution. Data were analyzed using student’s *t*-test and Mann Whitney U test, respectively, for the comparison of values between controls and whole NAFLD population. One-way ANOVA and Kruskal-Wallis test, respectively, were used to determine statistical differences between controls and patients stratified by stages of steatosis followed by Tukey’s (One-way ANOVA) or Dunn´s (Kruskal-Wallis test) post hoc test for multiple comparison. Differences in sex were analyzed using a chi-square test. All statistical analyses were performed using GraphPad Prism (version 7.03, 2017, GraphPad Software Inc., San Diego, CA, USA). A *p*-value ≤ 0.05 was considered significant.

### 2.6. Clinical Trial and Ethical Considerations

The Nutritional Counselling vs. Nutritional Supplements for NASH—a Randomized Prospective, Open Label Pilot Study (NUCES NASH study) was registered (clinical trials.gov NCT02366052) and approved by the ethics committee of the Landesaerztekammer Rheinland-Pfalz (Germany). The ENDO-META study (Effect of Monosaccharides on Intestinal Barrier Function) was registered (clinical trials.gov NCT 03482284) and approved by the ethics committee of the Medical University Vienna. The present study was performed in accordance with the ethical standards laid down in the 1964 Declaration of Helsinki and its later amendments. Written informed consent was given by all participants prior to study inclusion.

## 3. Results

### 3.1. Clinical Characteristics of the NAFLD Patient Cohort and Healthy Controls

The characteristics of NAFLD patients and age-matched controls are summarized in Appendix A. In NAFLD patients, body mass index (BMI) (23.3 vs. 31.5 kg/m^2^) and waist circumferences were significantly higher. Likewise, a significantly higher rate of metabolic comorbidities including higher prevalence of arterial hypertension, insulin resistance and hyperlipidemia were observed in patients with NAFLD (Appendix A). Interestingly, we observed a linear increase of BMI and waist circumference with increasing severity of hepatic steatosis (Figure 1a, Table 1). Paralleling the data found for BMI, levels of leptin were higher and levels of adiponectin significantly lower in patients with moderate and high degrees of hepatic steatosis (Figure 1b,c). Consequently, the leptin to adiponectin ratio—suggestive of increased visceral fat accumulation and adipose tissue dysfunction [21]—was significantly higher in these patients, too (Figure 1d). Standard laboratories separated patients with NAFLD from controls including significant increase in ALT, aspartate aminotransferase (AST) and gamma-glutamyl transferase (γGT) activity as well as fasting lipid profiles and uric acid concentration. Fasting blood glucose levels were significantly higher in NAFLD patients, with fasting insulin levels also being positively related to severity of steatosis (Table 1, steatosis grade 3 vs. controls: +23.2 mU/L, grade 3 vs. grade 2: +12.7 mU/L, grade 3 vs. grade 1: 16.4 mU/L, *p* < 0.05 for all comparisons). The metabolic comorbidities among the subgroups with different grades of hepatic steatosis was not significantly different—with the exception that a female predominance was observed in the high steatosis group. Metabolic and laboratory profiles were comparable (Table 1 and Appendix A).

### 3.2. Differences in Nutritional Intake in Patients with NAFLD and Healthy Controls

Interestingly, while dietary recalls did not capture significant differences in total intake of calories, fat, protein or carbohydrates between groups, we observed in general a significantly lower rate of dietary fiber intake in patients with NAFLD (Appendix A). The intake of fiber was lowest in patients with NAFLD suffering from grade 3 steatosis (17 g/day vs. 26 g/day in healthy controls, *p* < 0.05) while, despite being lower, the intake of fiber of patients with mild and moderate hepatic steatosis did not differ significantly from those of controls (grade 1 steatosis: 19g/day, grade 2 steatosis: 21 g/day) (Table 2).

### 3.3. Levels of Bacterial Endotoxin and LBP

Next, we examined levels of endotoxin and LBP in patients with NAFLD and controls. Compared to the control group, bacterial endotoxin levels in peripheral blood of all patients with NAFLD were almost double of those found in controls (0.66 ± 0.03 vs. 1.49 ± 0.12, *p* < 0.05) (Appendix A). When stratifying endotoxin levels according to severity of hepatic steatosis, patients with intermediate and severe hepatic steatosis exhibited significantly higher levels of bacterial endotoxin compared to healthy subjects (1.67 EU/mL and 1.62 EU/mL respectively vs. 0.66 EU/mL in healthy controls, *p* < 0.05 for both groups), while those of patients with mild hepatic steatosis were almost similar to controls (*p* = 0.086) (Figure 2). Interestingly, LBP concentration did not differ between controls and patients with NAFLD (LBP 25.1 ± 1.9 vs. 25.7 ± 1.7 [µg/mL]; Appendix A).

### 3.4. Relation of Hepatic Steatosis and Inflammatory Markers

Levels of systemic inflammation were assessed by active PAI-1 and CRP measurements. PAI-1 plasma levels were significantly higher when comparing all NAFLD patients to controls (+78%) (Appendix A). This was paralleled by significantly higher levels of high-sensitive CRP in patients with NAFLD. Interestingly, PAI-1 and CRP showed increasing levels when compared between the different levels of hepatic steatosis (Figure 3a,b).

## 4. Discussion

The current study explored nutritional intake in patients with NAFLD. The most striking difference observed between patients and controls regarding nutritional intake was related to the amounts of daily dietary fiber intake, however, we could not capture differences in the intake of total calories, fat, carbohydrates or proteins between controls and NAFLD. A role of fibers has been previously reported by others [15], also observing comparable caloric intake. Indeed, a higher dietary quality defined by fiber-rich foods including grains, cereals, vegetables, legumes and fruits has been suggested to decrease the risk of developing NAFLD [22]. In the literature, there is ample evidence that links dietary fiber to gut mobility and satiety—two aspects that could underlie the observed phenotype in patients with NAFLD as they regulate the risk to develop metabolic disease [23]. Intestinal mobility has also been linked to the intestinal permeability, a potential cofactor in disease progression of NAFLD in both animal models and humans [24]. Consequently, dietary interventions focusing on increasing dietary fiber content were able to demonstrate a decrease in surrogates of intestinal permeability in patients suffering from NAFLD [25].

We have previously shown that NAFLD predisposes to increased levels of endotoxin in children and adults with NAFLD [20,26,27] and these findings are again reflected in the current study. Importantly, we are able to recapitulate observations related to endotoxemia and link these to the dietary composition in a prospectively enrolling nutritional trial [27]. In contrast to endotoxin, no differences in LBP were observed in patients and healthy volunteers. In the literature, conflicting results have been reported [28] and it seems that while the measurement of LBP is technically less challenging, endotoxin measurement was more robust to capture differences in the current study. Nonetheless, the current study supports a role of endotoxin in NAFLD with intestinal permeability related to the amount of dietary fibers being one of the factors that could contribute to disease manifestation. More importantly, as one of the first studies, we have shown that both endotoxin plasma levels and dietary fiber intake are related to disease severity.

Metabolic inflammation is an emerging concept in patients with NAFLD to describe their multimorbidity. From a clinical perspective patients with NAFLD exhibit a significantly increased cardiovascular event risk [29], with NAFLD being an independent risk factor for non-fatal coronary heart disease and all-cause mortality with a hazard ratio (HR) of 1.43 after adjustment for traditional risk factors [30]. Animal models support the role of the hepatic compartment as one source of hepatic immune cell activation that links the liver to adipose tissue dysfunction [31]. In the current study, adipose tissue dysfunction is supported by increased levels of leptin and decreased adiponectin. Importantly, we also observed a correlation with the degree of non-invasively measured hepatic steatosis, thus providing additional evidence on the tight interaction between adipose tissue and the liver. The term metabolic inflammation is also increasingly used to explain emerging insulin resistance [32] and links the observations that are made with regards to increasing levels of fasting insulin, decreasing adiponectin, elevated PAI-1 and CRP—all findings aligned with the elevated levels of endotoxin. Thus, the association we observed in the analyzed cohort reflects very well the concept that metabolic inflammation is a common denominator of the observed clinical phenotype. According to our analysis, endotoxin—related to low dietary fiber content—could be one cofactor to account for these observations.

A second striking finding in our study was the observation that the severity of hepatic steatosis—when assessed by standardized ultrasound exam [15]—was associated with the degree of laboratory and clinical abnormalities in patients with NAFLD and importantly also with markers of inflammation. We observed incremental abnormalities with regards to the metabolic burden, comorbidities and increased liver function tests with increasing severity of hepatic steatosis. Interestingly, patients with mild hepatic steatosis were more closely resembling healthy controls than patients with moderate or advanced disease. This is interesting, considering that hepatic steatosis content is increasingly studied as a surrogate of disease improvement in early stage clinical trials—despite the fact that it was not previously shown to independently affect mortality [5]. Nonetheless, hepatic steatosis is a very useful clinical aspect for several reasons: (1) it can be assessed rapidly and non-invasively by many physicians without specialized training, (2) it predicts the presence and risk to develop metabolic comorbidities, e.g., diabetes type 2 or myocardial infarction [33,34] and (3) in the current study it correlated with LPS and markers of inflammation. Thus, the presence of moderate and severe hepatic steatosis could be a useful indicator to prioritize patients for dietary interventions, focusing on increasing dietary fiber content. Indeed, an increased dietary fiber intake has been related in several “beneficial” changes at the level of intestinal function including an increased diversity of microbiota, increased motility and synthesis of short chain fatty acids like butyrate, shown to be critical in modifying energy metabolism of enterocytes [35].

The study has several strengths and weaknesses. The sample size in particular of the control group constitutes a potential source of bias. We attempted to minimize this risk by matching of healthy controls to the patient group. The current observations are supported by previous studies on the link between intestinal permeability and low dietary fiber uptake in NAFLD. The robust differences in adipokine and endotoxin levels in relation to dietary fiber intake further support the validity of our findings. Additionally, we choose to classify patients non-invasively and thus we cannot adequately distinguish between NAFL and NASH or the presence of advanced fibrosis. However, the use of ultrasound offers the opportunity to transfer the findings outside of expert centers and thus can also be considered a potential strength of the analysis [36]. Secondly, underreporting during dietary assessment is of concern—even when performing standardized 24 h recalls by a trained nutritionist. Interestingly, we observed a difference with a high BMI despite low reported energy uptake, in particular in the steatosis S3 group. While the data cannot fully account for this discrepancy, it can be speculated that the contact with the study team has caused patients to adapt or report a lower calorie uptake. However, the robust differences in fiber intake are reassuring to support the validity of the applied nutritional assessment methods. One aspect when interpreting the lack of other nutritional factors to contribute to the observed phenotype—e.g., the amount of dietary fructose or total calories—is patient awareness that calories and sugar contribute to the metabolic phenotype, while this perception is not existent to the same extent for dietary fiber content and thus less vulnerable to this potential bias. Nonetheless, the baseline data reported here has several implications for the management of patients with NAFLD, which constitutes its central strength and relevance to clinical practice. From a clinician’s and patient’s point of view, it would be of highest interest to allow risk stratification and thus treatment allocation based on a simple, rapid and noninvasive measure such as an ultrasound. The association of hepatic steatosis with inflammation and an unfavorable metabolic phenotype, as well as the findings that dietary fiber content is linked to endotoxin in patients with moderate or advanced fibrosis, provides the opportunity to test a dietary intervention focusing on dietary fibers before performing liver biopsy to stage the patient and allow for experimental or pharmacological treatment.

## 5. Conclusions

In summary, the current study provides novel association on the degree of hepatic steatosis, inflammation and metabolic comorbidities—all related to the dietary fiber intake. While it is unlikely that dietary fiber is the single and most important aspect driving NAFLD, increased fiber uptake as the initial step in the management of patients with hepatic steatosis from NAFLD would provide a simple and easy to accomplish approach in the therapy of NAFLD patients. However, further studies are needed to systematically assess the impact of a fiber-enriched diet in the treatment of NAFLD, as well as related molecular mechanisms.

## Figures and Tables

**Figure 1 nutrients-12-00699-f001:**
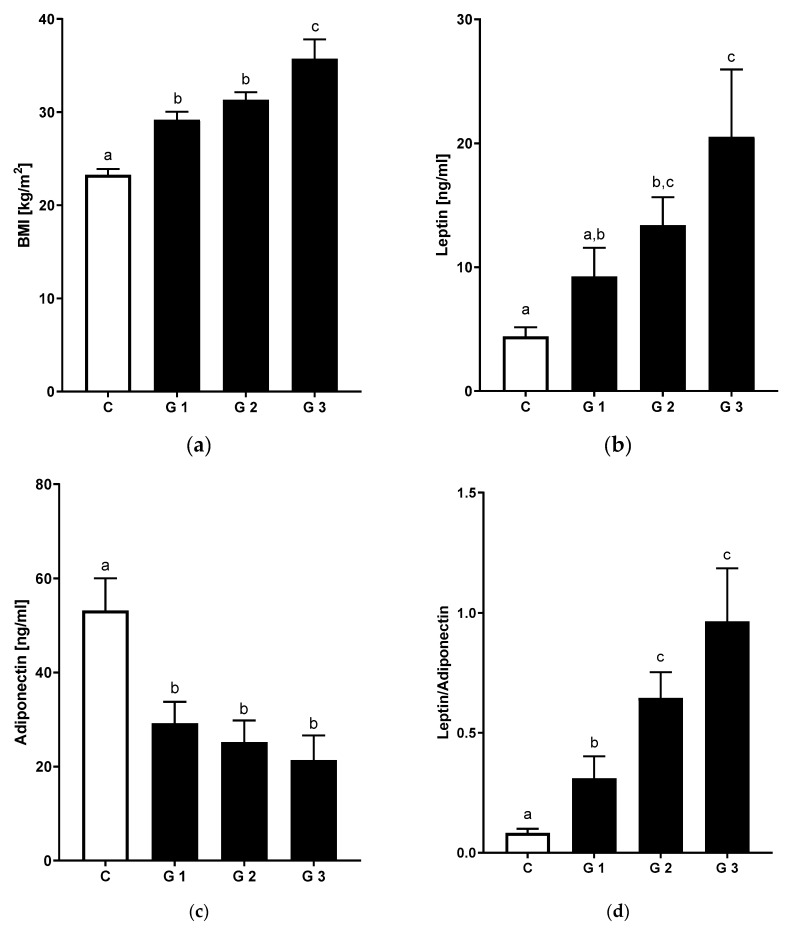
Metabolic parameters in non-alcoholic fatty liver disease (NAFLD) and healthy controls (**a**) BMI, (**b**) leptin and (**c**) adiponectin plasma levels as well as (**d**) leptin to adiponectin ratio in healthy controls and NAFLD patients by stages of steatosis. Data are shown as mean and standard error of mean (SEM). Different letters indicate significant differences between groups (*p* ≤ 0.05). BMI: body mass index. C: healthy controls; NAFLD with ultrasound-graded hepatic steatosis G1: mild; G2: moderate; G3: severe, leptin and adiponectin: analysis from patients with available blood samples (C: *n* = 14, G1: *n* = 13, G2: *n* = 19, G3: *n* = 6).

**Figure 2 nutrients-12-00699-f002:**
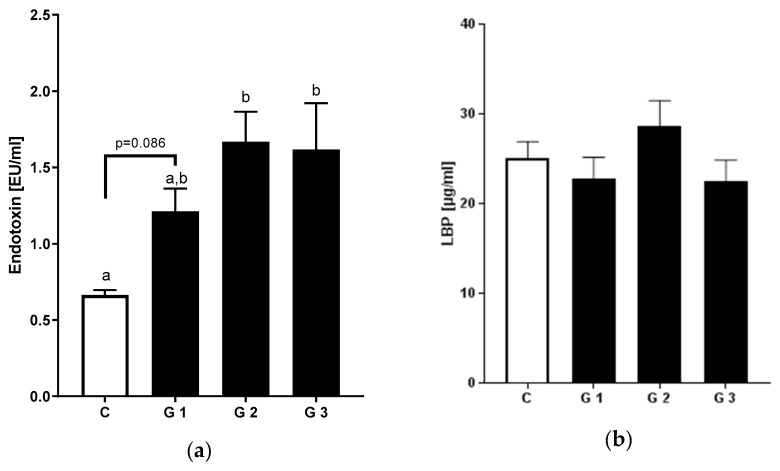
Markers of endotoxemia in NAFLD and healthy controls: (**a**) Endotoxin and (**b**) LBP plasma levels in healthy controls and NAFLD patients by stages of steatosis. Data are shown as mean and standard error of mean (SEM). Different letters indicate significant differences between groups (*p* ≤ 0.05). LBP: lipopolysaccharide binding protein. C: healthy controls; NAFLD with ultrasound-graded hepatic steatosis G1: mild; G2: moderate; G3: severe, analysis from patients with available blood samples (C: *n* = 14, G1: *n* = 13, G2: *n* = 19, G3: *n* = 6).

**Figure 3 nutrients-12-00699-f003:**
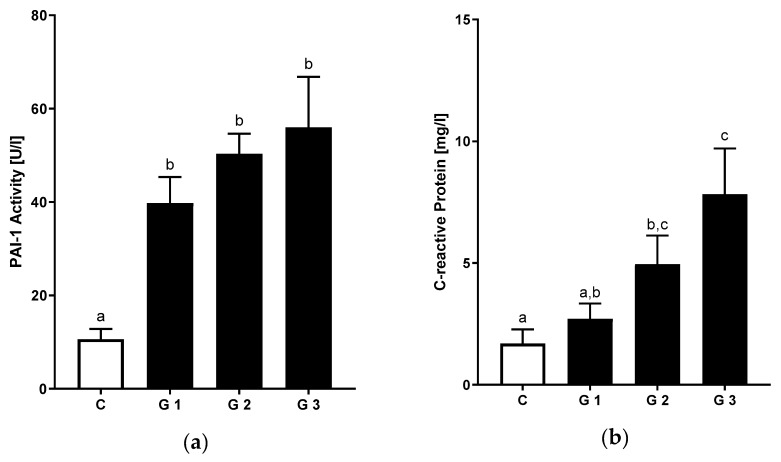
Markers of systemic inflammation in NAFLD and healthy controls: (**a**) PAI-1 activity in plasma and serum levels of (**b**) c-reactive protein in healthy controls and patients suffering from different stages of steatosis. Data are shown as mean and standard error of mean (SEM). Different letters indicate significant differences between groups (*p* ≤ 0.05). PAI-1: plasminogen activator inhibitor-1. C: healthy controls; NAFLD with ultrasound-graded hepatic steatosis G1: mild; G2: moderate; G3: severe, PAI-1: analysis from patients with available blood samples (C: n = 14, G1: n = 13, G2: n = 19, G3: n = 6).

**Table 1 nutrients-12-00699-t001:** Characteristics of healthy controls and patients with hepatic steatosis.

Parameter	Controls	Grade 1	Grade 2	Grade 3
n	14	20	31	12
Age	47.4 ± 1.2	51.5 ± 2.4	47.9 ± 2.4	52.2 ± 4.2
Sex [m/f]	4/10	11/9	20/11	2/10 ^$^
BMI [kg/m^2^]	23.3 ± 0.7 ^a^	29.2 ± 0.8 ^b^	31.3 ± 0.8 ^b^	35.7 ± 2.1 ^c^
Waist circumference [cm]	77.7 ± 2.3 ^a^	100.8 ± 2.8 ^b^	109.8 ± 2.2 ^b,c^	115.1 ± 4.3 ^c^
ALT activity [U/L]	18.4 ± 1.5 ^a^	58.7 ± 6.9 ^b^	82.6 ± 9.6 ^b^	59.4 ± 8.0 ^b^
AST activity [U/L]	18.4 ± 1.4 ^a^	38.2 ± 3.2 ^b^	47.4 ± 3.8 ^b^	54.3 ± 9.0 ^b^
γ-GT activity [U/L]	18.4 ± 2.3 ^a^	124.7 ± 22.4 ^b^	69.2 ± 8.5 ^b^	263.9 ± 100.5 ^b^
Systolic Blood Pressure [mmHg] ^#^	124.4 ± 2.6 ^a^	136.9 ± 3.8 ^a,b^	141.1 ± 3.6 ^b^	142.4 ± 5.6 ^b^
Diastolic Blood Pressure [mmHg] ^#^	82.3 ± 1.7	88.9 ± 2.4	88.1 ± 1.7	85.7 ± 2.6
Triglycerides [mg/dL]	87.0 ± 16.1 ^a^	147.5 ± 17.2 ^b^	174.2 ± 15.7 ^b^	193.1 ± 23.4 ^b^
Total Cholesterol [mg/dL]	195.6 ± 7.2 ^a^	232.1 ± 9.2 ^a,b^	208.5 ± 6.0 ^a,b^	240.6 ± 18.5 ^b^
HDL-Cholesterol [mg/dL]	72.9 ± 3.4 ^a^	55.6 ± 4.5 ^b^	43.7 ± 1.5 ^c^	51.1 ± 3.1 ^b,c^
LDL-Cholesterol [mg/dL]	107.2 ± 6.7 ^a^	147.0 ± 7.7 ^b^	129.4 ± 6.1 ^a,b^	150.8 ± 15.8 ^b^
Fasting Blood Glucose [mg/dL] ^#^	79.2 ± 2.8 ^a^	97.2 ± 2.3 ^b^	114.9 ± 5.8 ^b^	137.3 ± 18.5 ^b^
Fasting Insulin [mU/L] ^#^	4.4 ± 0.4 ^a^	11.2 ± 1.1 ^b^	14.9 ± 1.7 ^b^	27.6 ± 6.6 ^c^
HOMA-IR ^#^	0.9 ± 0.1 ^a^	2.8 ± 0.3 ^b^	4.6 ± 2.8 ^b^	10.6 ± 3.8 ^b^
Uric Acid [mg/dL]	4.2 ± 0.3 ^a^	6.0 ± 0.4 ^b^	6.3 ± 0.2 ^b^	6.2 ± 0.4 ^b^

ALT: alanine aminotransferase, AST: aspartate aminotransferase, γ-GT: γ-glutamyltransferase; Data are shown as total numbers or mean ± standard error of mean (SEM). Different letters indicate significant differences between groups (*p* ≤ 0.05). ^$^
*p* ≤ 0.05 in Fisher’s exact test. ^#^ missing data: blood pressure: *n* = 1, insulin: *n* = 5, fasting glucose: *n* = 1, HOMA: *n* = 6.

**Table 2 nutrients-12-00699-t002:** Nutritional intake of healthy controls and patients with steatosis.

Parameter	Controls	Grade 1	Grade 2	Grade 3
*n*	12 ^#^	19 ^#^	31	12
Total Energy [kcal/]	2194 ± 185	2167 ± 115	2184 ± 110	1860 ± 118
Protein [g/day]	82 ± 8	85 ± 6	83 ± 5	79 ± 7
Fat [g/day]	93 ± 10	96 ± 6	98 ± 7	73 ± 5
Carbohydrates [g/day]	242 ± 18	230 ± 18	233 ± 13	206 ± 23
Fiber [g/day]	26 ± 2 ^a^	19 ± 1 ^a,b^	21 ± 1 ^a,b^	17 ± 2 ^b^

Data are shown as total numbers or mean ± standard error of mean (SEM). Different letters indicate significant differences between groups (*p* ≤ 0.05). ^#^ 2 participants in the Control group and 1 participant in the steatosis grade 1 group (Grade 1) were excluded from the nutritional analysis due to incompliance in dietary assessment.

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
