# Peer review of "Adipokines and Endotoxemia Correlate with Hepatic Steatosis in Non-Alcoholic Fatty Liver Disease (NAFLD)"

_nutrients, 2020, doi:10.3390/nu12030699_

Round 1

Reviewer 1 Report

The manuscript entitled "Dietary Fiber Intake and Markers of Intestinal Permeability in Patients with NAFLD and Healthy Controls" by Nier et al conducted a prospective study to evaluate the markers of intestinal permeability in patients with non-alcoholic fatty liver disease (NAFLD), as well as the roles of macronutrient and fiber intake - as it may module intestinal barrier - in NAFLD patients versus healthy subjects.

The manuscript is well-written; abstract, introduction, results, and discussion sections are very well laid out whereas the materials and methods section provides comprehensive details of study protocol and methodologies, participants, and reagents. It is a very well-written report, such that authors have clearly pointed out the strengths and limitations of the study.

The present work provides limited evidence for the roles of nutritional fiber on NAFLD, which authors have also acknowledged. Additional well-controlled confirmatory studies are required to fully elucidate, and establish the roles of fiber-enriched diets on NAFLD as well as the underlying molecular mechanisms; the manuscript can be accepted for publication as it provides some evidence for fiber intake levels in patients with the degree of steatosis.

Author Response

We would like to thank the reviewer for his favorable response. We are in agreement with him that a well-controlled confirmatory studies is required to further elucidate and establish the roles of fiber-enriched diets on NAFLD as well as the underlying molecular mechanisms.

Reviewer 2 Report

This is a rigid clinical study to clarify the potential relationship between dietary fiber intake and prevention of NAFLD. All the results are as expected, rational and reliable. However, I have one critical concern.

Many preceding papers, such as Ref. 25 in the present manuscript, have had quite similar titles and conclusions to the present study. I recognize that the present study provided results simply supportive to the previous studies but not innovative. Hence, the authors should have more highlighted both concordant points and discordant points with the previous studies. This paper requires demonstrating clear distinction from the previous studies.

Author Response

The reviewer’s concerns related to overlap with existing studies and the need to be better highlight distinct findings from preceding papers are well recognized. We agree and have adapted title and conclusions to more rigidly reflect and highlight the novelty in the current study. Therefore, we have changed the title: Adipokines and endotoxemia correlate with hepatic steatosis in non-alcoholic fatty liver disease (NAFLD).

In addition, we have included a explanatory statements to out the finding into context in the following parts:

Lines 61-66: So far studies addressing endotoxin plasma levels in relation to dietary fiber intake and markers for visceral fat accumulation in patients with NAFLD are still lacking. […] Indeed, previous studies assessing intestinal permeability, abdominal fat accumulation and nutritional intake in patients with NAFLD did not stratify patients based on the degree of steatosis.

Lines 246-247: More importantly, for the first time we are able to show that both, endotoxin plasma levels and dietary fiber intake are related to disease severity in NAFLD.

Reviewer 3 Report

The manuscript of Nier A., et al.,  is a well written and presented manuscript aiming to assess differences among the NAFLD patients  belonging in different stages of the diseases and with healthy individuals. Introduction is well presented and centered on other relative studies. Language level is adecuate and the manuscript is easy to follow.

However, methodology has some serious problems. To begin with there is a great difference between number of control subjects and NAFLD patients which raises questions whether this difference has affected other outcomes of the study.  

Authors claim: "As liver histology was not available in all patients we use the term NAFLD to describe the study population throughout this manuscript." Does that mean that in the same group of NAFLD patients in stage 1 could be patients that belong in other stages but due to the absence of liver histology, the classification was not performed as it should? 

It is clear that patients were stratified in different stages of hepatic steatosis, according to well documented criteria. So, what about NAFLD patients without hepatic steatosis? Were there any? Furthermore, authors need to present the number and the particular caracteristics of the patients in the different groups (1, 2, 3).

Anthropometric caracteristics explain the differences between healthy subjects and patients. However, no differences in energy and macronutrient intake makes no sense with BMI and the evolution of the disease. Authors should consider whether this is due to the lower number of control subjects or the veracity of the 24h dietary recalls.  In case NAFLD patients have the same energy intake, this would mean that there are in process of losing body weght. Please consider possible causes for these results. 

Authors need to explain the statistical process they followed since the way they present it is confusing. Different letters should indicate statistical differences among the different groups and one way Anova would determine whether there are differences among the four experimental groups.

Discussion is well written and based on other relative studies. 

Author Response

Point 1: there is a great difference between number of control subjects and NAFLD patients which raises questions whether this difference has affected other outcomes of the study.

Response 1: The differences in the number of control and study subjects can be a limiting factor as the control group is more susceptible to outliers and we have included this limitation in the discussion. However by selecting age-matched, normal weight, healthy, non-smoking subjects with low alcohol intake we have minimized the risk of imbalance in the control group with regards to metabolic cofactors. As a matter of fact, it was a challenge to recruit a sufficient number of age-matched healthy volunteers. We introduced a statement to highlight these limitations (see lines 277-282).

Point 2: No liver histology was available Does that mean that in the same group of NAFLD patients in stage 1 could be patients that belong in other stages but due to the absence of liver histology, the classification was not performed as it should?

Response 2: We choose to classify patients based on the stage of hepatic steatosis by ultrasound. This is a reliable diagnostic approach and we added current literature to support this concept (Ref. 36 Hernaz et al.). However, in the absence of liver histology we cannot further explore the relationship of disease activity (inflammation; ballooning; NASH) or disease stage (hepatic fibrosis) in relation to our observations. This limitation was added (lines 284-287).

Point 3: what about NAFLD patients without hepatic steatosis? Were there any? Furthermore, authors need to present the number and the particular characteristics of the patients in the different groups (1, 2, 3).

Response 3: To better reflect the characteristics we amended table 1 and added further details of the different groups to a new figure 2. All patients included in the study exhibited hepatic steatosis according to I/E criteria, while the control group did not exhibit hepatic steatosis.

Point 4: Anthropometric characteristics explain the differences between healthy subjects and patients. However, no differences in energy and macronutrient intake makes no sense with BMI and the evolution of the disease. Authors should consider whether this is due to the lower number of control subjects or the veracity of the 24h dietary recalls. In case NAFLD patients have the same energy intake, this would mean that there are in process of losing body weight. Please consider possible causes for these results.

Reply 4: This point of the reviewer is well taken. We report the weight that was recorded at baseline and the nutritional data from baseline and during the 24 h recalls. Prior to enrollment weight was required to be stable in participants for 3 month. No data on physical activity is available. In this analysis, we can only speculate on the unaccounted differences between healthy controls and patients – which are reporting less energy intake – in particular in the S3 groups. These include (1) underreporting, (2) differences in physical activity and (3) changes that occurred between weight and 24 h recalls. In particular underreporting appears to be an aspect in the high BMI groups to adapt to a more “accepted” level of energy intake after the first contact during baseline. We included this limitation in the discussion. Despite potential underreporting of total calories, interestingly a low fiber content was still visible in the patient group supporting the central findings of our study. One explanation could be that while the patient could have underreported calories, the concept of a fiber rich diet as being “favorable” was not known to the patient and thus not stressed in the 24 h recall to the same extent (lines 289-292).

Point 5: Authors need to explain the statistical process they followed since the way they present it is confusing. Different letters should indicate statistical differences among the different groups and one way Anova would determine whether there are differences among the four experimental groups.

Response 5: We would like to thank the reviewer for highlighting this. We have amended the statistical analysis and description by including more in-depth description of the applied statistics and reorganized the figures (see lines 115-122).

Round 2

Reviewer 3 Report

Authors have improved the manuscript. No further comments